

# Quantifying the Cenozoic marine diatom deposition history: links to the C and Si cycles

Johan Renaudie[1]

[1]Museum für Naturkunde, Leibniz-Institut für Evolutions- und Biodiversitätsforschung, Berlin, Germany.

*Correspondence to:* Johan Renaudie (johan.renaudie@mfn-berlin.de)

**Abstract.** Marine planktonic diatoms are, today, among the world's main primary producers as well as the main organic carbon exporter to the deep-sea despite the fact that they were a very minor component of the plankton at the beginning of the Cenozoic. They are also the main silica exporter to the deep-sea, thus balancing global chemical weathering. This study reviews their global Cenozoic depositional pattern in order to understand the modality and the context of their rise to dominance, but also to understand how diatom evolution affected the Cenozoic functioning of the ocean's biological pump. After two short-lived major abundance peaks in the late Eocene and the late Oligocene, diatom abundance in sediments shifted in the mid-Miocene to globally higher values which have largely persisted to the modern day. These quantitative findings provide support for the hypothesis according to which diatoms, through their ecological role in the ocean's biological carbon pump, have contributed to the Cenozoic changes in atmospheric carbon dioxide pressure and consequently to changes in the global climate state. Additionally, correlations between diatom abundance peaks and shifts in seawater strontium and osmium isotopic composition hint at a strong control of the silicate weathering on diatom deposition.

## 1 Introduction

The Cenozoic history of oceanic silica deposition is a key piece of information in understanding the Cenozoic evolution of global weathering, ocean productivity and carbon accumulation, both as atmospheric $p_{CO_2}$ and as sedimentary $C_{org}$. Oceanic silica deposition, on scales of millions of years or more, is the main sink for silica released by continental and ocean crust weathering (Tréguer et al., 1995). Because dissolved ocean silica has a short residence time (Heath, 1974), marine opal deposition can be viewed as a proxy for the intensity of silicate weathering and cycling over time. In modern oceans, silica residence times are short in part because most of the silica is rapidly removed from ocean waters and sequestered as opaline sedimentary silica by marine planktonic diatoms (Heath, 1974; Nelson et al., 1995; Tréguer and Pondaven, 2000). This diatom production is also responsible for nearly half of ocean productivity (and approximately one quarter of global primary production), and is the single largest component of the ocean biologic carbon pump (Smetacek, 1999; Ragueneau et al., 2000, 2006). The history of silica deposition in Cenozoic oceans can thus also provide important insights into the history of the ocean's carbon pump (Smetacek, 1999; Cermeño et al., 2008). Marine planktonic diatoms, as the main driver of this process, are known to have themselves diversified dramatically over the Cenozoic (Lazarus et al., 2014), originating from primarily coastal, benthic ancestors in the late Cretaceous (Fenner, 1985). Many authors, in the absence of detailed records of Cenozoic opal deposition, have used



this diversity history of diatoms as a proxy for the intensity of opal export and deposition (Lazarus et al., 2014; Falkowski et al., 2004; Katz et al., 2007). Although diatom diversity and productivity are probably linked (Lazarus et al., 2014), the control of diatom diversity on the Cenozoic oceanic silica cycle and deposition of marine biogenic opal is still largely unexamined, as is the balance over the Cenozoic between diatom derived biogenic opal and that deposited by other organisms (mostly polycystine radiolarians and hexactinellid sponges).

Diatom primary productivity in the water column is usually limited by silicic acid availability in surface waters (Leynaert et al., 2001; Yool and Tyrrell, 2003; Brzezinski et al., 2011). It is concentrated in the modern ocean in several discrete areas of localised, wind-driven or regional upwelling of intermediate, more silica-rich ocean waters with at least seasonally weak water density stratification (Heath, 1974; Lisitzin, 1972; Garcia et al., 2010). Silica in the ocean is undersaturated at all depths (Garcia et al., 2010; Siever, 1957) (see also Fig. 1) and preservation as sediment only occurs when the export flux exceeds the dissolution rate in the water column and the upper layers of sediment (DeMaster, 2002; Tréguer and De La Rocha, 2013). Dissolution rates in the water column are in turn affected by patterns of deep-water flow between basin and deep ocean dissolved silica concentration (Berger, 1970). Accumulation of opal in marine sediments thus occurs only in a fairly complex, geographically patchy global pattern. It deposits primarily in the Southern Ocean belt, followed by the North Pacific and the Eastern Indian Ocean, as well as continental margin upwelling areas, and to a much lesser extent estuarine environment (Lisitzin, 1972; DeMaster, 1981) (Fig. 2). This makes reconstructing past global opal deposition more challenging than for carbonate, the other main biogenic mineral phase deposited in ocean sediments, since the saturation of water in respect to it depends primarily on pressure (i. e. depth) and is therefore largely constant across ocean basins: its deposition history can be thus reconstructed by a fairly small number of depth transects of sediment sections over time (Van Andel, 1975; Berger, 1978).

While abiotic removal of silica from the ocean is thought to have dominated until the late Cambrian (Maliva et al., 2005), since then, the oceanic silica cycle output have been almost exclusively biologically controlled (Maliva et al., 1989): until the Cretaceous, it seems to have been dominated by sponge spicules and radiolarians, and since then, by diatoms and radiolarians. Existing studies on the Cenozoic history of opal deposition (Muttoni and Kent, 2007; Baldauf and Barron, 1990; Miskell et al., 1985; Cortese et al., 2004; Brewster, 1980) are few, mostly qualitative, substantially limited in the data or methods used, and, for full Cenozoic scale, more than two decades old, thereby missing a great deal of new data from later drilling by ODP and IODP. A new synthesis is appropriate, and is the goal of this paper.

## 2   Material and Methods

The analyzed dataset is based on all smear slide descriptions published in the Initial Reports of the Deep-Sea Drilling Program (DSDP), Legs 1 to 96, Ocean Drilling Project (ODP), Legs 101 to 129 (available from National Geophysical Data Center, 2000, 2001) and Legs 178 to 201 (available from the Janus Database; Mithal and Becker, 2006), and Integrated Ocean Drilling Program (IODP), Legs 306 to 308 (also available from the Janus database). Legs 130 to 177 Initial Reports only contain semi-quantitative descriptions of smear slide components, and therefore were not used in this study. The omission of Legs 130-177 data does not seriously bias the results, although some smaller regions such as the Benguela Upwelling System, primarily



cored by Leg 175, are poorly represented in this analysis. The dataset consists, for each sample, of an estimate of the relative abundance of each elements seen in smear slides (microfossils or minerals). In total, the dataset contains 96669 samples, but only the 31136 samples for which a numerical age could be estimated with reasonable accuracy have been kept for the analysis. Out of the dated samples, 10832 contained diatoms, 9744 contained radiolarians, 3568 silicoflagellates and 4068

sponge spicules. While trends in smear slide and quantitative opal measurements tend to be very similar, smear slides do not provide estimates of chert abundance, and thus systematically underestimate opal in sections where significant silica diagenesis has occurred. This, however only affects significantly the older Paleogene record (Muttoni and Kent, 2007). The sources of the age models are detailed in Supplementary Table S1. All ages are given on the Gradstein et al. (2012) geomagnetic polarity time-scale. For the computations leading to Fig. 3 and 5, the data were binned into 1 Myr time intervals. The reason behind

that choice is twofold: high-resolution studies (Cortese et al., 2004) revealed patterns at the scale of 1 Myr; and, although the amount of data is sufficient when using 1 Myr steps to allow statistically significant results (see confidence intervals on Figs. 3 and 5) in most timebins, it is not the case when using narrower steps such as 0.5 Myr. Median relative abundance were calculated, as well as the interquartile range (i. e. from 1st to 3rd quartile; Tukey, 1977) and the 95% confidence interval on the median (McGill et al., 1978) for each 1 Myr time-bin.

The spatial pattern of the diatom distribution in the DSDP and ODP dated sediments is shown on Fig. 4 for each subepoch (from middle Eocene to Pleistocene). On those maps, the geographical pattern is interpolated between each samples available in said sub-epoch using Ordinary Kriging, based on an exponential semi-variogram model (Matheron, 1963; Cressie, 1993). The paleocoordinates of each DSDP and ODP sites, as well as the past continental configurations, were reconstructed using rotation poles, plates and coastlines datasets from the Earthbyte project (Müller et al., 2008). Paleogeographic reconstructions

were made using GPlates 1.4.0 (Boyden et al., 2011) and the interpolation and final maps were produced programmatically using R 3.2.3 (R Core Team, 2015). Similarly the spatial pattern of siliceous microfossils (diatoms as well as radiolarians, silicoflagellates and sponge spicules) in the Pleistocene was produced using the same method and is shown on Fig. 2, for comparison with the spatial pattern of biogenic opal measured in surface sediments, reconstructed from Archer (1996), and the spatial pattern of silicic acid at the bottom of the photic zone, reconstructed from Garcia et al. (2010).

## 25 3 Results

While radiolarians are blooming starting from ca. 50 Ma (early Eocene) until the earliest late Eocene (Muttoni and Kent, 2007; McGowran, 1989) (Fig. 3), the proportion of samples containing diatoms prior to 45 Ma (Middle Eocene) is too low to yield a reliable global median abundance. From that point on, however, diatom-bearing samples are numerous and geographically widespread, as they are found at all latitude and in all ocean basins (see Fig. 4) (Baldauf and Barron, 1990; Miskell et al.,

1985; Barron et al., 2015). At the transition between the late Eocene and the early Oligocene (from ca. 35 to 31 Ma), diatom abundance peaks (Fig. 3) in the high latitudes (Fig. 4), with a main deposition locus in the south Atlantic (Salamy and Zachos, 1999; Diekmann et al., 2004). Six Myr after this event, a smaller abundance peak occurs in the late Oligocene (between 26 and 24 Ma; Fig. 3). Once again, this event is more strongly marked in the southern high latitude, with, for the first time in the





Cenozoic, an higher abundance in all three sectors of the Southern Ocean (Fig. 4), hinting at a circum-Antarctic accumulation belt. During the early Miocene, diatom abundance in sediments is somewhat lower, with, for the first time, some loci of higher deposition in the middle latitude (Fig. 4), loci that will expand during the middle Miocene: this is most probably linked to the beginning of the coastal mid-latitude upwelling zones such as the Canary upwelling seen on the early and middle Miocene

maps. At ca. 15 Ma, the abundance pattern raises again progressively until it reaches a plateau of relatively high abundance (Fig. 3). The middle Miocene is also marked by a geographical change in the distribution of diatom abundance: until then, high abundances are limited to the Atlantic basin (and the Southern Ocean) while, starting from the mid-Miocene, the northern Pacific Ocean, followed later by the western Pacific and the eastern Indian Ocean dominate (Fig. 4). Direct records of biogenic opal for the Neogene (Baldauf and Barron, 1990; Cortese et al., 2004; Keller and Barron, 1983) confirm this shift in opal

deposition loci from the Atlantic (with noticeably a collapse of North atlantic opal) to the Pacific. Additionally, as was noted before (Brewster, 1980; Kennett, 1978), siliceous sedimentation becomes predominant in the Southern Ocean, thus establishing the modern Southern Ocean diatom belt (Burckle and Cirilli, 1987) (Fig. 4). The abundance plateau initiated during the middle Miocene event is more or less sustained until the Pleistocene, although the median relative abundance of diatoms seems to drop somewhat between 3 and 2 Ma (Fig. 3), yet still remaining above pre-15 Ma level. This drop is mainly affecting the low

to mid-latitudes, which might indicates a shrinking of the diatom polar-centered hotspots (Fig. 4) and reductions in abundance in the North Pacific. By contrast, the median abundance in the Indian Ocean seems to rise significantly (Fig. 4).

The spatial pattern observed for the Pleistocene for cumulated siliceous microfossils (Fig. 2) is largely coherent not only with the modern distribution of biogenic silica in sediments (Lisitzin, 1972; Archer, 1996) but also with the modern distribution of the silicic acid in the upper 200m layer of the ocean, i. e. the euphotic zone (Fig. 2) (Garcia et al., 2010): Spearman's rank

correlation coefficient (Spearman, 1904) on paired cells at $1°$ resolution yield $\rho = 0.464$ ($p < 2.2 \times 10^{-16}$) when comparing the smear slide data with Archer (1996) interpolated map, and $\rho = 0.399$ ($p < 2.2 \times 10^{-16}$) when compared with the silicic acid at 200m. Comparing with Fig. 4 shows that the main loci of diatom deposition in both the current analysis results for the Pleistocene and the reference datasets are the Southern Ocean, followed by the North Pacific and the eastern Indian Ocean while radiolarians add an opal deposition locus in the Pacific equatorial belt. Theses comparisons also show the adequacy of the

method to deliminate patterns of biogenic opal deposition, at least at broad scale, both temporally and spatially. The observed early Oligocene maxima in global diatom relative abundance is broadly coeval (Fig. 5) with an acceleration of the radiogenic strontium input (Barrera et al., 1991; Mead and Hodell, 1995; Zachos et al., 1999), with an abrupt shift in the $\delta^{18}O$ signal that indicates an abrupt cooling (Zachos et al., 2001, 2008), and a decrease of atmospheric $CO_2$ (Pagani et al., 2005). In addition to the strontium regime change, an abrupt drop in $^{187}Os/^{188}Os$ before the event (at 34.5 Ma), followed by an equally abrupt

increase (up to ca. 33.5 Ma) to finally reach a regime of slow increase (from ca. 33.5 to ca. 15 Ma), has also been described (Dalai et al., 2006). Both have been interpreted (Zachos et al., 1999; Dalai et al., 2006) as marks of increased weathering of the Antarctic continent following the early Oligocene glaciation (Oi-1). Similarly, the shift toward higher abundance of diatoms that starts at the beginning of the middle Miocene corresponds, temporally, to a shift in the oxygen isotope record – i. e. a cooling trend following the Mid-Miocene Climatic Optimum (Zachos et al., 2001) – and follows closely the rate of change

in the strontium isotope record (Hodell et al., 1991; Hodell and Woodruff, 1994; Ravizza and Zachos, 2003). In addition to



the strontium record, the osmium isotope record also underwent a clear break in the early middle Miocene, changing from an Oligocene to Miocene plateau to a rapid late Neogene increase (Peucker-Ehrenbrink et al., 1995). The coupling of those two isotope systems (Sr and Os) have been interpreted as an indication of intense continental weathering, namely of the Himalayas (Peucker-Ehrenbrink et al., 1995).

The shift in diatom relative abundance observed near the Eocene/Oligocene boundary (and already noted in the literature (Baldauf and Barron, 1990; Katz et al., 2004; Kooistra et al., 2007) is also accompanied by a coincident, substantial increase in diatom diversity (Lazarus et al., 2014) as well as an increase in provincialism (Lazarus et al., 2014; Fenner, 1985). Not only does the Southern Atlantic seem to be the main focus of the late Eocene - early Oligocene event, but the Southern Ocean flora is known to have undergone a strong turnover at the Eocene-Oligocene transition (Kennett, 1978; Baldauf, 1992). The mid-
Miocene shift in diatom abundance pattern is also associated with a well-documented diatom diversification event (Lazarus et al., 2014) (Fig. 5). Lazarus et al. (2014) showed that the vast majority of the living species of marine diatoms originate from 15 Ma onwards.

## 4    Discussion

Although diatom depositional history mimics neither the $^{87}Sr/^{86}Sr$ record nor the $^{187}Os/^{188}Os$ record, it does seem to react
noticeably to changes in long-term rates in these parameters (in the earlier Oligocene and the early middle Miocene), suggesting significant control of rates of weathering on Cenozoic diatom abundance: increased weathering resulting in increased silica availability in the world's oceans and thus increased export of silica by diatoms to the sedimentary record. Diatom export is also linked to Cenozoic climate history since both major shifts in weathering rates are temporally correlated to the two largest Cenozoic cooling events: Oi-1 (Zachos et al., 2001) and the middle Miocene climatic transition (Flower and Kennett, 1994).
Inferring the direction of causal relationship is difficult here because of the low temporal resolution of Figs 2–5, however silicon isotopes ($\delta^{30}Si$) measurements (Egan et al., 2013) on late Eocene - early Oligocene antarctic diatoms and sponges showed that increased diatom productivity (through an increase in silicic acid utilisation) started before the Oi-1 event – the first high values of $\delta^{30}Si$ occurring as early as ca. 37 Ma, and increasing until 34.5 Ma, which seem to concur with the increase in diatom abundance seen here as starting in the 36 to 35 Ma time bin.

The largest single fraction of the global Cenozoic signal is from changes in the southern high latitudes (Fig. 4). Diatom started deposited in all sectors of the Southern Ocean as early as the late Oligocene although it did not form a proper diatom accumulation belt before the middle Miocene. A distinct Southern Ocean surface circulation with endemic biotas extends back to the late Eocene (Lazarus et al., 2008). The Antarctic Circumpolar Current (ACC) with its current physical properties (specifically its depth) however may have formed only in the late Oligocene (at ca. 25 Ma Lyle et al., 2007), though there is
still debate on the subject (Barker and Thomas, 2004). If a late Oligocene formation of the ACC is confirmed, then, considering the timing and the geographical extent of the late Oligocene diatom accumulation event, it is plausible that this new circulation pattern triggered increased opal deposition in the Southern Ocean. The shift of the main diatom deposition locus starting at ca. 15 Ma from the Atlantic to the Pacific oceans (the 'silica switch') is one of the main changes in geographic pattern seen





in the Cenozoic history of silica deposition. The reasons for this switch have been interpreted by several previous authors (Baldauf and Barron, 1990; Cortese et al., 2004; Keller and Barron, 1983; Flower and Kennett, 1994) as a consequence of the formation of the North Atlantic Deep Water component which suppressed transport and upwelling of Antarctic Bottom Water into the North Atlantic, thus turning the Atlantic Ocean into a 'lagoonal'-type ocean (sensu Berger (1970), i. e. with deep-water

outflow) and leading to silica-richer waters in the Pacific and Indian Ocean ('estuarine'-type sensu Berger (1970), i. e. with deep-water inflow).

Past fluctuations of species diversity and links to possible causal factors, e.g climatic or environmental changes, are central themes in paleobiology. Most scenarii invoked to explain such correlations implicitly assume that diversity is strongly corre-lated to ecologic abundance, even though this correlation is rarely tested. Although there is no strict one-to-one relationship

between Cenozoic diatom abundance and diversity (Fig. 5), the primary periods of diversity change and turnover (Lazarus et al., 2014; Cervato and Burckle, 2003) occurred during periods of abundance shifts (the ca. 33 Ma and the ca. 15 Ma events). Only the Late Oligocene abundance maximum does not correspond to any known global diversity increase. This may be ex-plainable by the relatively narrow geographical extent of the event (Fig. 4). There is additionally in the Neogene a strong correlation between the abundance curve presented herein and the diversity curve (Lazarus et al., 2014): the correlation over

the last 25 Myr has a Pearson's $r = 0.77$ ($p = 9.8 \times 10^{-6}$); and a Pearson's $r = 0.52$ ($p = 0.009$) when detrended linearly.

Biogenic opal accumulation rates (which in the Cenozoic in most regions corresponds almost exclusively to the diatom accumulation rate (Tréguer and De La Rocha (2013); Fig. 2) are often used as an indicator of paleoproductivity (Hüneke and Henrich, 2011). The studied dataset only gives access to diatom relative abundance in the sediment (not absolute abundance, or accumulation rate) so the variations seen here do not correspond to absolute variations of the paleoproductivity but, at best,

to variations of the diatom contribution to the paleoproductivity. However, the diatom contribution to primary production is known to be (today) the dominant exporter of carbon to the deep sea, i. e. this is the main component of the ocean's biological pump (Smetacek, 1999; Lazarus et al., 2014); diatom relative abundance in the sediments should therefore indicate the relative strength of the biological carbon pump versus the contribution of the so-called 'alkalinity pump', that corresponds mainly to the activity of calcareous nannoplankton (Frankignoulle et al., 1994). A comparison of diatom abundance with the $p_{CO_2}$

Cenozoic curve (Beerling and Royer, 2011) (Fig. 5) show that each increase of diatom abundance corresponds to a drop in $p_{CO_2}$: at ca. 33 Ma, ca. 26 Ma and ca. 15 Ma. Higher global diatom relative abundance does seem to be linked with decreased atmospheric $p_{CO_2}$, meaning not only that the diatom relative abundance might indeed be a good proxy for export productivity but also that there is evidence in our results for a diatom-to-climate causal interaction during most of the Cenozoic as has been speculated or modeled before (Tréguer et al., 1995; Lazarus et al., 2014; Falkowski et al., 2004; Katz et al., 2007; Pollock,

1997). The use of 1 Myr time bins is of course still too broad to show directly a cause-and-effect relationship, but the observation of a clear correlation and a coherent model to explain it (i. e. the biological carbon pump) are already strong evidence in favor of such a relationship. The resulting scenarii for each of the two main events observed, should this relationship be correct, would thus be the following. At the mid-Miocene event, the uplift of the Tibetan Plateau causes increased silicate weathering, which in turn favors an increase in diatom abundance in the oceans; consequently, the atmospheric $p_{CO_2}$ decrease which,

together with the various feedback mechanisms resulting directly from the Himalayan orogenesis (Raymo and Ruddiman,



1992), induces a global cooling. The Eocene-Oligocene transition event and interactions between $p_{CO_2}$ and polar circulation have been extensively discussed in the literature, but, if the timing of events implied by Egan et al. (2013) is correct, the role of diatoms is to provide the $p_{CO_2}$ forcing that results in Antarctic continental glaciation: the opening of the Drake Passage allows a surface circum-Antarctic current to circulate, thus bringing more nutrients to the southern Atlantic diatom communities,

increased diatom productivity, long-term carbon export into marine sediments, drawdown of atmospheric $p_{CO_2}$ and a cooling event. This cooling, together with the thermic isolation of the Antarctic continent by the proto-ACC, results in the formation of the East Antarctic Ice Sheet; the resulting weathering of the Antarctic continent by this ice-sheet creating finally a positive feedback through an even higher input of silica into the oceans and higher diatom abundance.

When comparing the variations of total biogenic opal (i. e. diatoms, radiolarians, silicoflagellates and sponge spicules)

in deep-sea sediments during the Cenozoic, with variations in diatom and radiolarian abundance in sediments (Fig. 3), one striking feature is that, prior to the Eocene-Oligocene boundary, the total biogenic opal variations is almost exclusively due to radiolarians but that from the Oligocene onwards, it is the diatom which accounts for most of the biogenic opal variation. Harper and Knoll (1975) suggested the possibility of a competition – on a geological time scale – between diatoms and radiolarians for dissolved silica in sea water. Later, Lazarus et al. (2009) confirmed the reduction in radiolarian silica use, and showed

that it primarily resulted from a marked trend in tropical radiolarian shell thinning around the Eocene-Oligocene transition, with little change seen in polar regions. Lazarus et al. (2009) argued that the tropical restriction of this morphologic trend was evidence for strong, diatom-dominated silica removal from low latitude surface waters, beginning in strongly stratified post-Eocene oceans. They were not able to compare their radiolarian data to any direct measures of opal export, but showed that the Cenozoic pattern of decreasing radiolarian silica use matched that of increasing diatom diversity. Similarly, a model-

based approach by Cermeño et al. (2015) confirmed the impact of an increased silica flux on the dominance of diatoms over radiolarians, through competition. This take-over of the biological marine Si cycle by diatoms at this time is here confirmed.

The deposition of biogenic opal is the only output from the marine Si cycle (Tréguer et al., 1995), while continental weathering is the dominant input to this system, with only smaller contributions from hydrothermal activity, aeolian dust and seafloor weathering. In theory, biogenic opal deposition should therefore compensate global silicate weathering on a geo-

logical timescale. The curve presented here (Fig. 3) shows only relative abundances of biogenic silica-bearing microfossils in sediments, not accumulation rates; however, unless there have been systematic changes in average global total biogenic sediment accumulation rates on Myr scales over the Cenozoic, the globally averaged relative abundance data presented here should largely track variations in global opal accumulation rates, and could therefore, in theory, be used as a proxy for Cenozoic silica weathering rate.

**5   Summary and Conclusions**

Diatoms have increased in abundance over the Cenozoic, with two main abundance events, one at the Eocene-Oligocene transition, another during the Mid-Miocene. These events correlate with shifts in seawater strontium and osmium isotope composition, hinting at a strong control of the weathering on diatom abundance. Paleoceanographic and paleoclimatic events



such as the onset of the Antarctic Circumpolar current and the establishment of a permanent Northern Hemisphere Ice Sheet might have also exerted control on local events of diatom abundance (respectively the Southern Ocean late Oligocene maxima and the collapse of the North Pacific diatom deposition in the late Pliocene).

Beyond the simple abundance shift, the Mid-Miocene event also witnessed a complete spatial reorganization of the diatom deposition loci, switching from the Atlantic to the Pacific and Indian basins. Mid-latitude upwelling zones also appeared during this event, and it is also during this event that the modern Southern Ocean diatom accumulation belt formed.

The findings presented here also provide support for the scenario in which diatoms, through their ecological role in the ocean's carbon pump, might be responsible at least in part for Cenozoic changes in atmospheric carbon dioxide pressure and therefore changes in global climate state. The quality and spatiotemporal coverage of the studied data is not yet sufficient however to determine the sequence of changes, in diatom abundance versus climate changes, and therefore causality cannot be determined with any statistical certainty. Similarly, the role of diatom in the Si cycle is shown here to be dominant since the Eocene-Oligocene transition, hinting that the abundance of diatoms in sediments might echo quantitatively the amount of global chemical weathering.

*Author contributions.* J.R. conceived and conducted the study, analyzed the results and wrote the manuscript.

*Acknowledgements.* The author is currently supported financially by a DFG (German science foundation) grant RE 3470/1-3. The author would like to thank D. Lazarus for his support during this study and J. Witkowski and M. Schobben for constructive feedback on previous versions of this manuscript.



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



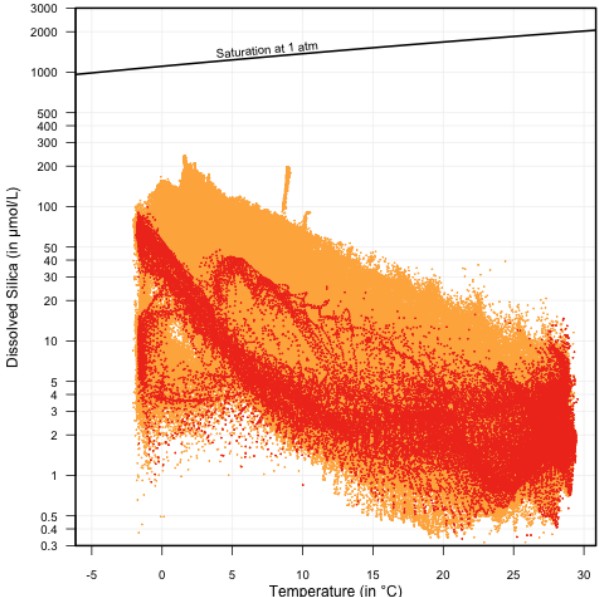

**Figure 1.** Dissolved silica is undersaturated at all depths. Saturation concentration vis-à-vis amorphous silica at 1 atm of pressure follows Wollast (1974). It is known to increase with pressure/depth (e. g. Willey, 1974). Red dots corresponds to measurements of dissolved silica (silicic acid) at the surface of the ocean and orange dots at various depths. Both corresponds to the complete World Ocean Atlas 2009 dataset (Garcia et al., 2010).

Zachos, J., Pagani, M., Sloan, L., Thomas, E., and Billups, K.: Trends, rhythms, and aberrations in global climate 65 Ma to present, Science, 292, 686–693, 2001.

Zachos, J. C., Opdyke, B. N., Quinn, T. M., Jones, C. E., and Halliday, A. N.: Early cenozoic glaciation, antarctic weathering, and seawater $^{87}$Sr/$^{86}$Sr: is there a link?, Chemical Geology, 161, 165–180, 1999.

5   Zachos, J. C., Dickens, G. R., and Zeebe, R. E.: An early Cenozoic perspective on greenhouse warming and carbon-cycle dynamics, Nature, 451, 279–283, 2008.



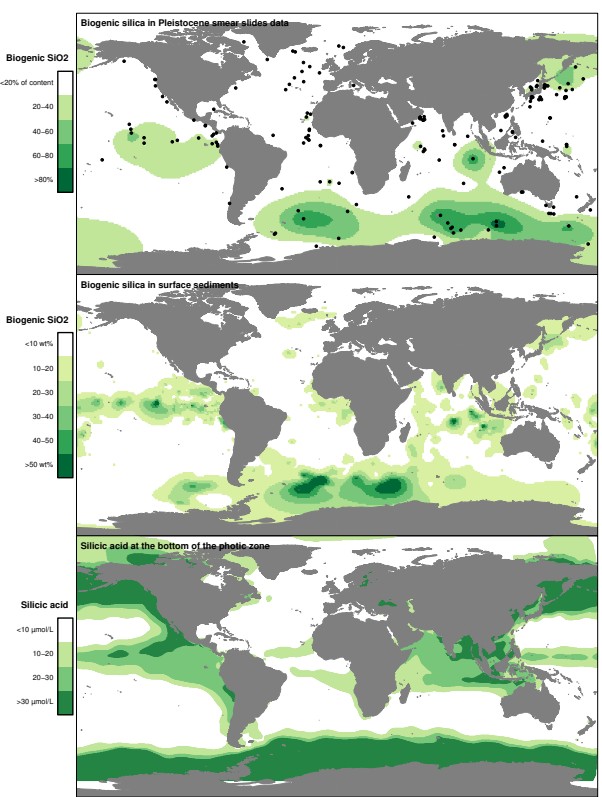

**Figure 2.** Comparison between smear slides-based data and reference data. Upper panel: cumulated distribution of diatoms, radiolarians, silicoflagellates and sponge spicules in the Pleistocene. Black dots are the siliceous microfossil-bearing sites present in the Pleistocene. Middle panel: biogenic opal in surface sediments from Archer (1996). Lower panel: silicic acid at 200m below the ocean's surface (lower limit of the euphotic zone) from Garcia et al. (2010).





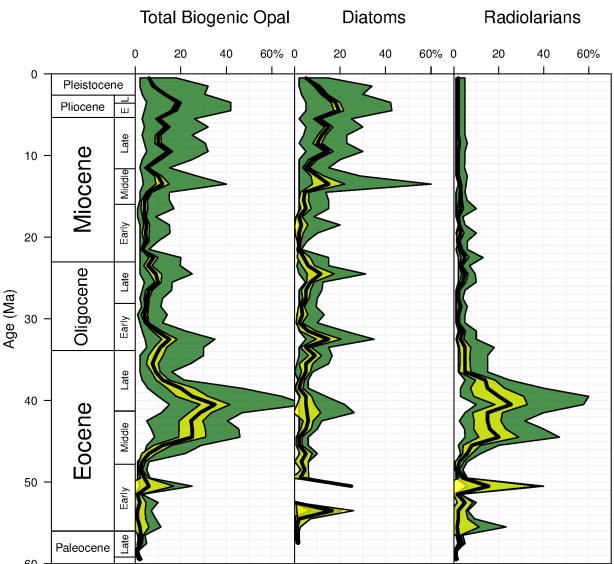

**Figure 3.** History of biogenic opal deposition. Left panel: total biogenic opal in DSDP-ODP smear slides (diatoms, radiolarians, silicoflagellates and sponge spicules combined). Middle panel: global abundance of diatoms. Right panel: global abundance of radiolarians. Bold black line is the median value in each 1 Myr time-bin while the yellow envelope delimits the 95% confidence interval on the median and the green enveloppe the interquartile range of the data.




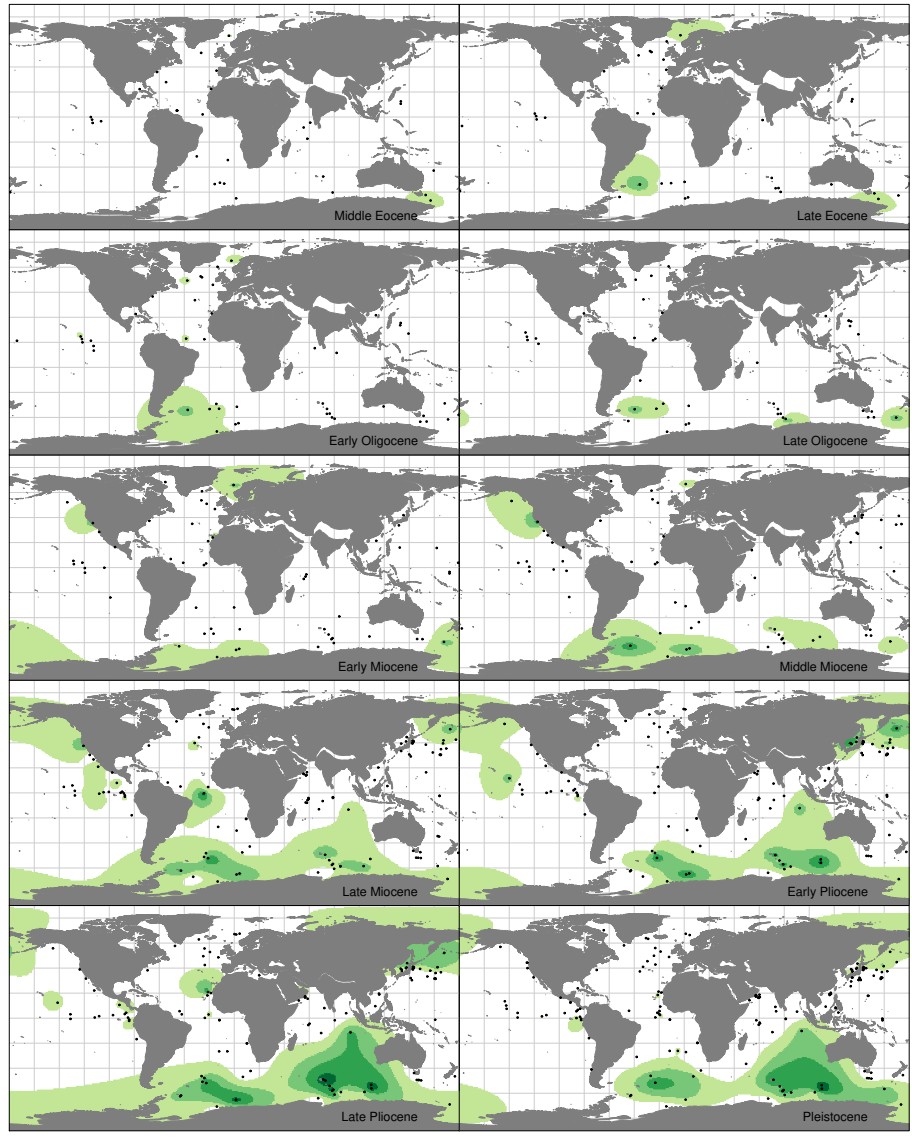

**Figure 4.** Maps of diatom abundance in DSDP-ODP smear slides during the Cenozoic. Black dots are the diatom-bearing sites present in the concerned sub-epoch. From white to dark green, the categories corresponds to: less than 20%, 20 to 40%, 40 to 60%, 60 to 80%, more than 80% (see Fig. 2 for color scale). Diatom abundance is interpolated using ordinary kriging based on an exponential model. Paleogeography from Müller et al. (2008).





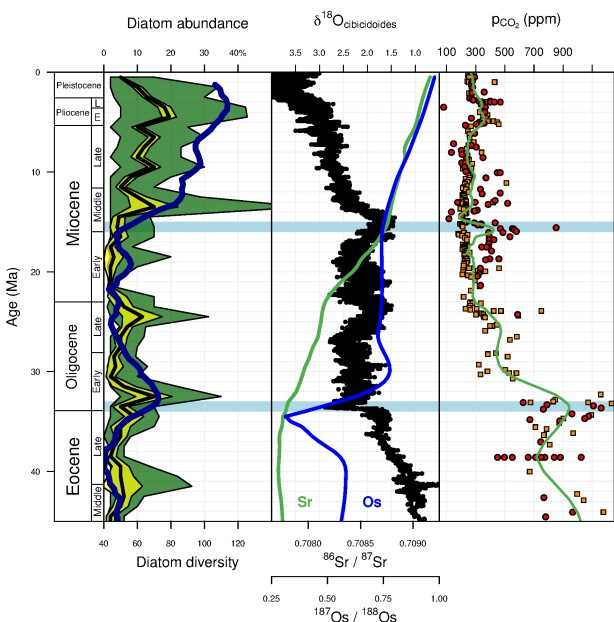

**Figure 5.** Comparison with climate and oceanography proxies. First panel – diatoms. Bold blue line is the diatom diversity from Lazarus et al. (2014). Diatom abundance as in Fig. 3. Second panel – benthic $\delta^{18}O$ (black) from Zachos et al. (2008), LOESS regression curve (Cleveland et al., 1992) of strontium (green) and osmium (blue) isotope data based on compilation by Ravizza and Zachos (2003). Third panel – Cenozoic $p_{CO_2}$ reconstruction. Orange squares (phytoplankton-based proxy) and red dots (other proxies) from Beerling and Royer (2011). Green line is a LOESS regression curve of this data.