# Peer review of "Quantifying the Cenozoic marine diatom deposition history: links to the $\mathbf C$ and $\mathbf S\mathbf i$ cycles"

_Biogeosciences, 2016_

## Referee Comment (RC1) · Anonymous Referee #1 · 11 Aug 2016

General considerations This paper corresponds to an updated synthesis on the Cenozoic marine diatom deposition history that that compiles existing ODP and IODP data. It is an important paper with global scientific significance and quality that certainly belongs in the scope of BG. There are, however, some general and specific aspects that should be taken into account before the paper can be accepted for publication. The main problems are, in my view, the non-consideration of ODP legs that took place on the major coastal upwelling systems and that should be included in the database, such as Leg 202 on the SE Pacific or Leg175 on NW Africa. Without including these data, no clear conclusions can be reached for the history of the mid-latitude upwelling zones. Os and Sr are mentioned as weathering proxies, but there is no clear explanation as to why this is the case and that needs to be included in the introduction. The silica switch is mentioned on line 200 of pg. 6, but there is no explanation or even reference to it. You cannot assume that all the readers know about it, so at least a short explanation and a reference is necessary. Again, on line 242 of pg. 7, you consider the Himalayan orogenesis and resulting feedback mechanisms, which together with increased diatom production were the cause for global cooling, however you do not mention those mechanisms and that needs to be specified, as well as their contribution to global cooling. From the Late Miocene to the Present, variability in the total diatom abundance, although minor than at the large events discussed in the text, does not show any correspondence to pCO2 decreases. Shouldn't the reason for that be explained or at least hypothesized?

Specific aspects Specific aspects relate to the structure of the paper. The author has decided to separate the results from the discussion but ends up discussing also in the results section. It might be better to merge the two in a results and discussion section with subheadings to guide the reader through the text. Besides, the presentation of results and discussion should follow a chronological order, from the oldest to the younger subepochs. Also please associate ages to the considered geologic periods throughout the text. Presentation of references within the text needs to follow an order, either by alphabetic name of author, or by year of publication and this needs to be consistent throughout the text. In terms of language there is a need for revision in some parts, in order to improve the fluency of the text and provide an easier reading of the MS. Examples of paragraphs that need revision are: lines 120 and 125 in pg 4; lines 142 to 148 in pg 5; lines 182 to 189 in pg 6; lines 248 to 253 in pg 8; lines 255 to 261 in pg 8; Figure 4 – The size of the maps is so small that the paleogeographic differences between subepochs are difficult to see.

---

## Referee Comment (RC2) · Anonymous Referee #2 · 18 Aug 2016

This paper provides an important review of the constraints on marine diatom deposition in the modern ocean and evaluates diatom distribution through time using smear slide estimates compiled from the DSDP and most of the ODP cruises. The maps produced are very useful in understanding the spatial distribution of marine diatoms through time and their patterns are mostly supported by the literature. However, there are fundamental problems with the assumptions of using this data. Shipboard sedimentologists who have variable experience in the recognition of microfossils traditionally compile smear slide abundance data. Smear slide preparation is not uniform and is subject to bias based on the preparation technique. Estimation of the relative abundance of various microfossils can be quite different depending on the magnification at which the smear slide is examined. For example, the most abundant diatoms typically are very small compared to radiolarians and may be missed at magnifications of X250 or less.

[Figure]

It is acknowledged on line 25 of p. 7 that the smear slide data presented does not represent accumulation rates: however, there is no justification for the statement that that "globally averaged smear slide data should largely track variations in global accumulation rates". In most of the sediments recovered from tropical ocean DSDP-ODP sites, as well as those from the Atlantic Ocean as a whole, calcium carbonate is most abundant biogenic sediment component observed in smear slides. Treatment of these samples in hydrochloric acid reveals abundant diatoms. Such acid treatment has been the subject of numerous papers on diatom biostratigraphy published in the DSDP-ODP literature. Similarly, along active margins, such as the eastern coasts of North America, diatoms are masked in smear slide abundance by detrital materials. In both cases, biostratigraphic and quantitative studies have revealed quite different diatom abundance patterns than those shown on Figure 4. Baldauf and Barron (1990) and Barron et al 2015), which are both cited in this paper, take this into consideration and present quite different results for the Eocene. Compilation of accumulation rates requires knowledge of the rates of sediment accumulation.

Specific points – 1) The abstract mentions a diatom abundance peak in the late Eocene, but Figures 3 and 4 show that this peak is actually just above the Eocene/Oligocene boundary. 2) Figure 4 suggests that late Oligocene increase in smears slide diatom abundance is limited to the Southern Ocean. I suspect that in may reflect a decrease in calcium carbonate abundance south of the Antarctic Polar Front. 3) The reader should be aware of a 2016 PNAS (v. 113, no. 25) paper by Crampton et al., which gives a thorough discussion of the response of Southern Ocean diatoms to $CO_2$ changes during the past 15 million years.

---

## Referee Comment (RC3) · R. Scherer (Referee) · 23 Aug 2016

By compiling visually estimated microfossil and biogenic silica estimates based on a very large data set, this paper provides a significant contribution to our understanding of oceanic silica production over time, and by implication continental weathering since the Cretaceous. But by focusing on weathering, the author may be underplaying the impact of unrelated open ocean upwelling over time.

I have read the assessments of reviewers 1 and 2 and strongly agree with all of their independent assessments of the strengths and weaknesses of the paper. I especially endorse Reviewer 2's comments about the uncertainties associated with shipboard smear slide analysis. Many parameters conspire to reduce the recognition of diatoms in shipboard smear slides: (1) carbonate obscures diatoms, (2) Slides have been pre-

pared using different mounting media – some are fairly low refractive index, making diatoms difficult to resolve, (3) partial dissolution of silica dramatically impacts smear slide assessment, (4) slides made by the lithostratigraphy team are often still made on microscope slides rather than on the cover glass, making it impossible to analyze slides at high magnification, and (5) slides are often only analyzed at low magnification, leaving diatoms unrecognized. The most significant variable is the varied experience of the team. It's been my experience that visual estimates of diatom content can vary dramatically based on the parameters outlined above. The impact of these significant uncertainties is lessened by the large data set, so I am confident that the overall trends described are real. The manuscript will be improved with these caveats discussed.

P3/Ln 7. There are many examples of post-Paleogene cherts, including Quaternary examples. Such early diagenesis would greatly underestimate diatom abundance, as would relatively low abundance due to dilution from high terrigenous input.

P4/ln 24: "the main loci of diatom deposition in both the current analysis results for the Pleistocene and the reference datasets are the Southern Ocean, followed by the North Pacific and the eastern Indian Ocean while radiolarians add an opal deposition locus in the Pacific equatorial belt." Some of this is undoubtedly higher dissolution among diatoms vs rads. This is in part a direct result of the rads that live beneath sinking diatoms. The paper generally lacks a discussion of the impact of silica dissolution, which is often dramatic and will skew the interpretation. Silica dissolution is a complex process that regionally alters the paleoproductivity signal. (Some new papers discuss the record and potential impact of diatom dissolution; e.g., recent papers by Warnock).

Some additional editorial comments regarding usage throughout the paper. 1. Although technically acceptable, there is heavy reliance on parenthetical comments and clauses. In many cases this complicates rather than simplifies the content.

2. P5/Ln 25-26: Diatoms accumulated in abundance in all sectors. . .

3. Stratigraphic usage: The author of this paper does a pretty good job of avoiding

common errors with regard to time and time-rock units (Late vs Upper, etc.), but there are a few items that I take issue with. (a) When used as formal stratigraphic terms, Upper, Lower and Middle should be capitalized – doing this helps distinguish formal stratigraphic assignment. For example, "Upper Miocene" refers to a specific, defined stratigraphic subunit, whereas "upper Miocene" less specifically refers to rocks deposited during the later part of the Miocene. (b) There are also some misuses of tense in stratigraphic description. Bear in mind the difference between time and rocks (e. g., T. rex lived during the Late Cretaceous, but T. rex fossils occur in Upper Cretaceous rocks). The author makes some quite minor errors of tense in a few places. For example, P3/Lns 31 and 32 should be past tense (peaked and occurred, rather than peaks and occurs) because he is referring to an event in time (which is past) rather than the geological manifestation (which still exists in the present). Other examples include (1) P4/Lns 2, 3: During the Early Miocene, diatom abundance in sediments was somewhat lower. . . loci that expanded during the Middle Miocene and (2) P4/Lns 14 + 16: This drop mainly affected. . ..in the Indian Ocean seems to have risen significantly.

4. Typos, etc: 2/20 Although rather than While. 4/5 rises rather than raises. 4/7 whereas instead of while. 4/10 Atlantic. 4/24 These. 4/25 delineate? 8/11 diatoms. There is some mixing of UK and US English usage.

My overall assessment is that this paper is very worthy of publication with the relatively minor changes recommended by me and my fellow reviewers.

---

## Author Comment (AC1) · 5 Oct 2016

Thanks to Anonymous Referee #1 for their thoughtful review.

Concerning the absence of the Humboldt and the Benguela mid-latitude upwelling zone from the data: as mentioned in the Material and Methods section, leg 175 is not included in the dataset because, at that time, the smear slides descriptions were given semi-quantitatively instead of quantitatively making it difficult to integrate these data in the study; as for Humboldt, while leg 202 is present in the dataset, only Pleistocene sediments were recovered in the four sites of leg 202 that were drilled in the upwelling zone, thus preventing the reconstruction for the Miocene and Pliocene. Please note however that both the Canary (Leg 41) and the California (Leg 63) mid-latitude up-welling zones are present in the dataset. I will include in the revisions however men-

tions to articles that treated the middle to late Miocene appearance of the Benguela upwelling zone in particular, as they were currently missing in the manuscript. I will also include more background informations on the use of the Os and Sr isotopes, on the Silica Switch and on the direct impact of weathering on the climate. Concerning the structure of the paper, indeed I agree that merging Results and Discussions while separating subsections by time interval could improve the readability of the manuscript. Concerning the size of the maps: while in the latex-produced submission file Figure 4 appeared small it is in fact meant as a full page figure.

---

## Author Comment (AC2) · 5 Oct 2016

Thanks to Anonymous Referee #2 for their thoughtful review.

Concerning the quality of the smear slide descriptions: indeed I will add a more thorough discussion on all the caveats concerning smear slide descriptions in the Material and Methods, as, of course, I am aware of them. I would like however to point out that Figure 2 shows that, despite all their problems, smear slide descriptions data do seem to preserve correctly the spatial pattern (though clearly at a lower resolution). What was missing however are figures showing that the per-site temporal trends are also preserved in these data: I attach here a figure (that I could add as supplementary figure if needed) comparing directly biogenic silica abundance as seen through the lens of the smear slide descriptions and actual measurement of biogenic silica using chemical methods (namely density separation, double leaching by Na2CO3 and wet alkaline extraction; data from Bohrmann 1988, Gurvich 1988, Forehlich et al 1991 and Wang et al. 2004) on a few sites for which both data are available on a long enough depth range to enable direct visual comparisons. They do show that, even if there is not an exact 1-to-1 correspondence (which is not to be expected anyway as one measures a percentage of objects when the other measures a weight percentage), the temporal trend is indeed preserved.

Concerning the statement that "globally averaged smear slide data should largely track variations in global accumulation rates", I was merely pointing out the idea that, unless there has been secular changes in global sedimentation rates, trends in relative abundances and in accumulation rates shouldn't differ widely from one another when averaged globally. I could modify this sentence however if this is seen as being too speculative.

Specific Points: 1) While the peak itself of the higher abundance event is indeed slightly above the E/O boundary (though bear in mind that the resolution is 1-Myr), the beginning of this event is slightly below the E/O boundary which is why i referred to it as being Late Eocene. I'll try to clarify this in-text. On a related note, a reader made me realized I incorrectly used the base of the Bartonian instead of the base of the Priabonian as base of the late Eocene: it will be corrected in the final Figure. 2) The fact that the event seems limited to the Southern Ocean is discussed in-text. Concerning the depletion of carbonates, while this is indeed a possibility (and I will mention it in the revised text), I think carbonate microfossils (judging by the smear slides data) are still fairly abundant in the Southern Ocean up to the late Miocene (see for instance fig. 6B of Renaudie & Lazarus 2013).

Additional references:

Bohrmann, Gerhard (1988): Zur Sedimentationsgeschichte von biogenem Opal im nördlichen Nordatlantik und dem Europäischen Nordmeer. Berichte aus dem Sonderforschungsbereich 313, Christian-Albrechts-Universität, Kiel, 9, 221 pp (dataset: http://dx.doi.org/10.1594/PANGAEA.79531)

Froelich, Philip N; Malone, PN; Hodell, David A; Ciesielski, Paul F; Warnke, Detlef A; Westall, Francis; Hailwood, Ernie A; Nobes, DC; Fenner, Juliane M; Mienert, Jürgen; Mwenifumbo, CJ; Müller, Daniel W (1991): Biogenic opal and carbonate accumulation rates in the subantarctic South Atlantic: The late Neogene of Meteor Rise Site 704. In: Ciesielski, PF; Kristoffersen, Y; et al. (eds.), Proceedings of the Ocean Drilling Program, Scientific Results, College Station, TX (Ocean Drilling Program), 114, 515-550 (dataset: http://dx.doi.org/10.1594/PANGAEA.754799)

Gurvich, Evgeny G (1998): Metallonosnye Osadki Mirovogo Okeana (Metalliferous Sediments of the World Ocean). Nauchnii Mir (Moscow), 340 pp (dataset: http://dx.doi.org/10.1594/PANGAEA.773679)

Renaudie, Johan and Lazarus, David (2013): On the accuracy of paleodiversity reconstructions: a case study in antarctic Neogene Radiolarians. Paleobiology, 39(3), 491-509.

Wang, Rujian; Li, Jianru; Li, Baohua (2004): Data report: Late Miocene-Quaternary biogenic opal accumulation at ODP Site 1143, southern South China Sea. In: Prell, WL; Wang, P; Blum, P; Rea, DK; Clemens, SC (eds.) Proceedings of the Ocean Drilling Program, Scientific Results, College Station, TX (Ocean Drilling Program), 184, 1-12. (dataset: http://dx.doi.org/10.1594/PANGAEA.785059)

[Figure]

**Fig. 1.** Comparison between smear slide-derived (blue) and geochemically-mesured biogenic silica (red) measurements for DSDP site 408 and ODP sites 704A and 1143A.

---

## Author Comment (AC3) · 5 Oct 2016

Thanks to Reed Scherer for his thoughtful review.

See reply to Anonymous Referee #2 concerning the quality of the smear slide descriptions. Concerning the post-Paleogene cherts: indeed there are some cherts present in post-Eocene sediments, however as Muttoni & Kent 2007 (Fig. 2) showed, their abundance starting from the middle Eocene onwards (after horizon Ac) is considerably lower than before and therefore unlikely to affect significantly the manuscript results in my opinion.

Concerning the comment on page 4 line 24: indeed there is a differential in dissolution rate between diatoms and radiolarians that lowers the abundance of diatom in Equatorial Pacific sediments (as it was discussed in particular in Lisitzin 1972, due to the

effect of temperature on the kinetics of the dissolution). I will add some comments concerning the dissolution issue in the manuscript. Please note however that this doesn't affect the conclusions concerning the silica cycle as dissolved diatoms are de facto not part of the output of the marine silica cycle.

Thanks also for the many comments concerning the language used in the text: i will indeed edit the text as much as possible to follow these recommendations.
* * *

---

## Author Response (AR1)

Following the associate editor notes and the reviewers comments, I modified the text to include caveats regarding the use of smear slides data (together with an additional Supplementary Figure showing how temporal trend are nonetheless conserved on a site-by-site basis) and a more precise explanation concerning the chert issue. Some sentences have been reworked to be more legible, or to offer clearer explanations. Follows a point-by-point response to the reviews, and a file showing the modifications done (using latexdiff).

Johan Renaudie.

**Anonymous Reviewer #1:**

* "The main problems are, in my view, the non-consideration of ODP legs that took place on the major coastal upwelling systems and that should be included in the database, such as Leg 202 on the SE Pacific or Leg175 on NW Africa."

> This has been answered in my public reply to this comment. In short, leg 175 is absent from the dataset (as is said in Material & Methods) and leg 202 only recovered Pleistocene sediments. Since indeed not all mid-latitude upwelling zones are discussed in text, I got rid of the sentence fragment mentionning them in the conclusions.

* "Os and Sr are mentioned as weathering proxies, but there is no clear explanation as to why this is the case and that needs to be included in the introduction."

> There is quite a wide literature on the subject (the relevant part of which is cited here in-text). The simple explanation on why they are used as weathering proxy is because the isotopic ratio is different in the continental and in the oceanic plate, so changes in the ratio means changes in the continental input, in the form of chemical erosion (to simplify).
> As I do not interpret those signals myself (but instead rely on existing, and cited, interpretations), I do not feel it is necessary for this manuscript to include such explanation.

* "The silica switch is mentioned on line 200 of pg. 6, but there is no explanation or references to it"

> References have been added more explicitly. Explanations were already given implicitely: the event known as the Silica Switch is the shift in deposition locus of biogenic silica from the Atlantic to the Pacific. I modified slightly the sentence nonetheless to make it explicit.

* "on line 242 of pg. 7, you consider the Himalayan orogenesis and resulting feedback mechanisms, which together with increased diatom production were the cause for global cooling, however you do not mention those mechanisms and that needs to be specified, as well as their contribution to global cooling."

> The two main feedbacks were added explicitly. The full list can be found in the cited article.

* "From the Late Miocene to the Present, variability in the total diatom abundance, although minor than at the large events discussed in the text, does not show any correspondence to pCO2 decreases."

> As mentioned by the other reviewers, they are reasons to believe smear slides data are not perfectly reliable. In this paper I showed that the main broad-scale trends are in fact reliable, which is why I analyze large scale pattern and large temporal variation (typically the 3 main

events I discuss correspond to 10-20% increases in global relative abundance). Though it would be tempting to analyze smaller variations, I do not have strong evidence that they are reliable, and not just noise due to the data source.

* "Presentation of references within the text needs to follow an order, either by alphabetic name of author, or by year of publication and this needs to be consistent throughout the text."

Being new to LaTeX, i thought the references would be ordered automatically by year. They were not and this is now corrected.

* "Figure 4 – The size of the maps is so small that the paleogeographic differences between subepochs are difficult to see."

Figure 4 is meant to be a full page figure, in the published version of the manuscript.

**Anonymous Reviewer #2:**

* "there are fundamental problems with the assumptions of using this data [...]"

This has been addressed by the addition of a small text explaining some of the issues with smear slides data and by the addition of supplementary figure 2.

* "It is acknowledged on line 25 of p. 7 that the smear slide data presented does not represent accumulation rates: however, there is no justification for the statement that that "globally averaged smear slide data should largely track variations in global accumulation rates"."

This was addressed in my public reply to this comment. This is, in text, explicitly said in the context of the assumption according to which they were no secular, global change in sedimentation rates.

* "1) The abstract mentions a diatom abundance peak in the late Eocene, but Figures 3 and 4 show that this peak is actually just above the Eocene/Oligocene boundary."

As discussed in my public reply to this comment, the peak of this event is indeed slightly above the E/O boundary (though keep in mind that the resolution is only 1Myr) but the start of the event is just below. I modified nonetheless the abstract from 'in the late Eocene and late Oligocene' to 'near the Eocene/Oligocene boundary and in the late Oligocene' to avoid confusion.

* "2) Figure 4 suggests that late Oligocene increase in smears slide diatom abundance is limited to the Southern Ocean. I suspect that in may reflect a decrease in calcium carbonate abundance south of the Antarctic Polar Front."

The geographical limit of the late Oligocene event was discussed in-text already. I added a small sentence about the possibility of it being linked to a decrease in carbonates (following my public reply to this comment).

**Reed Scherer:**

* "Many parameters conspire to reduce the recognition of diatoms in shipboard smear slides"

See response to reviewer #2

* "There are many examples of post-Paleogene cherts"

Following what I mentioned in the public response to this comment, I modified the sentence in the material and methods section that was already discussing cherts.

* "The paper generally lacks a discussion of the impact of silica dissolution, which is often dramatic and will skew the interpretation"

Though silica dissolution is indeed an important subject and affect indeed diatoms more than radiolarians, diatoms that get dissolved before being buried in the sediment do not participate to the output of the marine silicon (or indeed carbon) cycle and therefore should not affect the conclusions of this study, in my opinion. Regarding partial dissolution, it would be a problem indeed if I was using weight percentages, instead of elemental percentages. I added a sentence however, when discussing the Equatorial Pacific Belt, to mention the differential dissolution of diatoms vs radiolarians as a probable cause.

* "1. Although technically acceptable, there is heavy reliance on parenthetical comments and clauses."

I tried to correct the main offenders.

* "2. P5/Ln 25-26: Diatoms accumulated in abundance in all sectors"

Done.

* "3. Stratigraphic usage"

I corrected the errors in tense usage.

* "4. Typos"

All corrected.

[revised manuscript text omitted]